# Fast Certified Robust Training with Short Warmup

Zhouxing Shi [* 1]   Yihan Wang [* 1]   Huan Zhang [1 2]   Jinfeng Yi [3]   Cho-Jui Hsieh [1]

## Abstract

State-of-the-art (SOTA) methods for certified robust training including interval bound propagation (IBP) and CROWN-IBP usually use a long warmup schedule with hundreds or thousands epochs and are thus costly. In this paper, we identify two important issues, namely exploded bounds at initialization, and the imbalance in ReLU activation states, which make certified training difficult and unstable, and thereby long warmup was previously needed. For fast training with short warmup, we propose three improvements, including a weight initialization for IBP training, fully adding Batch Normalization (BN), and regularization during warmup to tighten certified bounds and balance ReLU activation states. With a short warmup for fast training, we are already able to outperform literature SOTA trained with hundreds or thousands epochs under the same network architecture.

## 1. Introduction

Methods for improving the empirical adversarial robustness of DNNs, such as adversarial training (Madry et al., 2018), provide no provable robustness guarantees, while some recent works aim to pursue *certified robustness*. Specifically, the robustness is evaluated in a certifiable manner using robustness verification methods (Katz et al., 2017; Zhang et al., 2018; Wong & Kolter, 2018; Singh et al., 2018; 2019; Bunel et al., 2017; Raghunathan et al., 2018b; Wang et al., 2018b), which verify whether the model is provably robust under all possible input perturbations, usually by efficiently computing the output bounds.

To improve certified robustness, *certified robust training* methods minimize a certified loss computed by a verifier, and the certified loss is an upper bound of the worst-case

---

*Equal contribution [1]UCLA [2]CMU [3]JD.COM. Correspondence to: Zhouxing Shi <zshi@cs.ucla.edu>, Yihan Wang <yihanwang@cs.ucla.edu>.

*Accepted by the ICML 2021 workshop on A Blessing in Disguise: The Prospects and Perils of Adversarial Machine Learning.* Copyright 2021 by the author(s).

loss given specified input perturbations. So far, Interval Bound Propagation (IBP) (Gowal et al., 2018; Mirman et al., 2018) and CROWN-IBP (Zhang et al., 2020; Xu et al., 2020) are the most efficient and effective methods for general models. IBP computes an interval with the output lower and upper bounds for each neuron, and CROWN-IBP further combines IBP with tighter linear relaxation-based bounds (Zhang et al., 2018; Singh et al., 2019) during warmup. They both can have a per-batch training time complexity similar to standard DNN training. However, certified robust training remains costly, mainly due to their need for a long warmup schedule where the perturbation radius for training is gradually increased from 0 to the target value. For example, generalized CROWN-IBP in Xu et al. (2020) used 900 epochs for warmup and 2,000 epochs for a CNN model on CIFAR-10 (Krizhevsky et al., 2009).

In this paper, we identify two important issues in existing certified training. First, we find that the certified bounds can be exploded at the training start, which is partly due to the suboptimal *weight initialization*. Prior works generally use weight initialization originally designed for regular network training, while certified training is essentially optimizing an augmented network (Zhang et al., 2020). Existing initializations can lead to exploded certified bounds at initialization. The long warmup with gradually increasing perturbation radii in prior works can somewhat be viewed as finding a better initialization for final training with the target radius, but it is too costly. Second, we also observe that *IBP leads to imbalanced ReLU activation states*, where the model prefers inactive (dead) ReLU neurons significantly more than other states because inactive neurons tend to tighten IBP bounds. It can however hamper classification performance if too many neurons are dead.

We focus on improving IBP training since IBP is efficient per batch, with the following improvements: **1)** We derive a new weight initialization, *IBP initialization*, for IBP-based certified training. The new initialization can stabilize the tightness of certified bounds at initialization; **2)** We identify the benefit of Batch Normalization (BN) in certified training, and we find BN can balance ReLU activation states and also stabilize variance. We propose to fully add BN to every layer, while it was partly or fully missed in prior works; **3)** We further propose regularizers to explicitly stabilize certified bounds and balance ReLU activation states.

We are able to efficiently train certifiably robust models that outperform previous SOTA performance in significantly shorter training epochs. We achieve a verified error of **65.03%** ($\epsilon = \frac{8}{255}$) on CIFAR-10 in **160** total training epochs, and **82.36%** on TinyImageNet ($\epsilon = \frac{1}{255}$) in **80** epochs, based on efficient IBP training. Under the same convolution-based architecture, we significantly reduce the total training cost by $20 \sim 60$ times compared to previous SOTA (Zhang et al., 2020; Xu et al., 2020) or concurrent work (Lyu et al., 2021).

## 2. Background

The objective of robust optimization is: $\min_\theta \mathbb{E}_{(\mathbf{x},y) \in \mathcal{X}} \left[ \max_{\delta \in \Delta(\mathbf{x})} L(f_\theta(\mathbf{x} + \delta), y) \right]$, for neural network $f_\theta$, data $\mathbf{x}$, ground-truth $y$, perturbation $\delta$ constrained by $\Delta(\mathbf{x})$, and loss function $L$. *Adversarial training* (Goodfellow et al., 2015; Madry et al., 2018) solve the inner maximization with adversarial attack. For robustness guarantees, *certified robust training* computes a certified upper bound for the inner maximization. Some works used costly relations (Raghunathan et al., 2018a; Wong & Kolter, 2018; Mirman et al., 2018; Dvijotham et al., 2018; Wang et al., 2018a)., while Interval Bound Propagation (IBP) (Mirman et al., 2018; Gowal et al., 2018) computing interval bounds is an effective and more efficient method. CROWN-IBP (Zhang et al., 2020) further combined IBP with linear relaxation bounds (Zhang et al., 2018) during warmup, generalized and accelerated in Xu et al. (2020). In concurrent works, Lyu et al. (2021) proposed a parameterized ramp function for activation; Zhang et al. (2021) proposed to use a different architecture with "$\ell_\infty$ distance neurons". Yet these works still need long training schedules.

## 3. Methodology

### 3.1. Notations and Definitions

We focus on improving IBP training. We consider a commonly adopted $\ell_\infty$ perturbation setting in adversarial robustness on a $K$-way classification task. For a DNN $f_\theta(\mathbf{x})$ with clean input $\mathbf{x}$, there can be some perturbation $\delta$ satisfying $\|\delta\|_\infty \leq \epsilon$, and the actual perturbed input to the model is $\mathbf{x} + \delta$. We want to verify whether

$$[f_\theta(\mathbf{x} + \delta)]_y - [f_\theta(\mathbf{x} + \delta)]_i > 0, \quad \forall \|\delta\|_\infty \leq \epsilon, i \neq y \quad (1)$$

holds true, where $[f_\theta(\mathbf{x}+\delta)]_i$ is the logit score for class $i$ and $y$ is the ground-truth. This is equivalent to verifying whether the DNN provably makes correct prediction for all input $\mathbf{x}+\delta$ ($\|\delta\|_\infty \leq \epsilon$). We assume that there are $m$ hidden affine layers (either convolutional or fully-connected layers) with ReLU activation. We use $\mathbf{h}_i$ to denote the pre-activation output value of the $i$-th layer, and use $\mathbf{z}_i = \text{ReLU}(\mathbf{h}_i)$ for

post-activation value. We use $\mathbf{W}_i$ and $\mathbf{b}_i$ to denote the parameters of the convolutional or fully-connected layer, where $\mathbf{W}_i \in \mathbb{R}^{r_i \times n_i}, \mathbf{b} \in \mathbb{R}^{r_i}$, and $r_i$ and $n_i$ are called the "fan-out" and "fan-in" number of the layer respectively (He et al., 2015b). In particular, we use $\mathbf{h}_0 = \mathbf{x} + \delta$ to denote the input layer and $\mathbf{z}_0$ is not applicable. IBP computes and propagates the lower and upper bound interval of each $\mathbf{h}_i$ layer by layer until the last year or verification objective, denoted as interval $[\underline{\mathbf{h}}_i, \overline{\mathbf{h}}_i]$ such that $\underline{\mathbf{h}}_i \leq \mathbf{h}_i \leq \overline{\mathbf{h}}_i$ ($\forall \|\delta\|_\infty \leq \epsilon$). Finally Eq. (1) can be verified by checking the lower bound of $[f_\theta(\mathbf{x} + \delta)]_y - [f_\theta(\mathbf{x} + \delta)]_i$.

### 3.2. Issues in Existing Certified Robust Training

In this section, we analyze two issues in existing certified robust training methods.

**Exploded Bounds at Initialization** For affine layer $\mathbf{h}_i = \mathbf{W}_i \mathbf{z}_{i-1} + \mathbf{b}_i$, IBP computes:

$$\underline{\mathbf{h}}_i = \mathbf{W}_{i,+}\underline{\mathbf{z}}_{i-1} + \mathbf{W}_{i,-}\overline{\mathbf{z}}_{i-1} + \mathbf{b}_i, \quad (2)$$

$$\overline{\mathbf{h}}_i = \mathbf{W}_{i,+}\overline{\mathbf{z}}_{i-1} + \mathbf{W}_{i,-}\underline{\mathbf{z}}_{i-1} + \mathbf{b}_i, \quad (3)$$

where $\mathbf{W}_{i,+}$ stands for retaining positive elements in $\mathbf{W}_i$, and vice versa for $\mathbf{W}_{i,-}$. We check the tightness: $\Delta_i = \overline{\mathbf{h}}_i - \underline{\mathbf{h}}_i = |\mathbf{W}_i|(\overline{\mathbf{z}}_{i-1} - \underline{\mathbf{z}}_{i-1}) = |\mathbf{W}_i|\delta_{i-1}$, where $\Delta_i$ denotes the difference between the upper and lower bounds, which can reflect the tightness of the bounds, and $|\mathbf{W}_i|$ stands for taking the absolute value element-wise. We assume each $\mathbf{W}_i$ is randomly initialized with each weight following a distribution with zero mean and variance $\sigma_i^2$. Then we view $\Delta_i$ as a random variable and use $\mathbb{E}(\Delta_i)$ to measure the expected tightness at layer $i$. As $\mathbf{W}_i$ and $\delta_{i-1}$ are independent, we have $\mathbb{E}(\Delta_i) = n_i \mathbb{E}(|\mathbf{W}_i|)\mathbb{E}(\delta_{i-1})$. Detailed in Appendix D.1, we further have $\mathbb{E}(\delta_i) = \mathbb{E}(\text{ReLU}(\overline{\mathbf{h}}_i) - \text{ReLU}(\underline{\mathbf{h}}_i)) = \frac{1}{2}\mathbb{E}(\Delta_i)$, and

$$\mathbb{E}(\Delta_i) = \frac{n_i}{2}\mathbb{E}(|\mathbf{W}_i|)\mathbb{E}(\Delta_{i-1}). \quad (4)$$

**Definition 1.** *We define the difference gain when bounds are propagated from layer $i - 1$ to layer $i$:*

$$\mathbb{E}(\Delta_i)/\mathbb{E}(\Delta_{i-1}) = \frac{n_i}{2}\mathbb{E}(|\mathbf{W}_i|). \quad (5)$$

*Bounds are considered to be stable if the difference gain* $\mathbb{E}(\Delta_i)/\mathbb{E}(\Delta_{i-1})$ *is close to 1.*

A large difference gain indicates explosion, but it cannot be much smaller than 1 either to avoid signal vanishing in the model. We find that weight initialization in prior works have large difference gain values especially for layers with larger $n_i$. For example, for the widely used Xavier initialization (Glorot & Bengio, 2010), the difference gain is $\frac{1}{4}\sqrt{n_i}$, and it can be as large as 45.25 when $n_i = 32768$ for a fully-connected layer in experiments. This indicates that certified

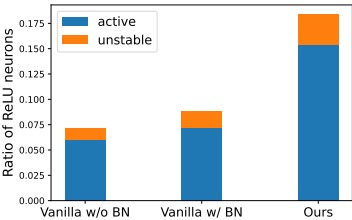

*Figure 1.* Ratios of active and unstable ReLU neurons a CNN on CIFAR-10 with different settings. The vanilla ones do not have regularization, and "vanilla (w/o BN)" does not use BN either.

bounds are exploded at initialization. We illustrate the explosion and compare different initializations in Appendix A. As a result, small perturbation radii are used in the early stage of the training to gradually make the model suitable for the target perturbation radius, but it is inefficient.

**Imbalanced ReLU Activation States** We also find another issue where the models have a bias towards *inactive ReLU neurons*, defined as neurons with non-positive pre-activation upper bounds ($\overline{\mathbf{h}}_{i,j} \leq 0$ for some neuron $j$ in layer $i$).Similarly, *active ReLU neurons* have non-negative pre-activation lower bounds.There are also *unstable ReLU neurons* with uncertain activation states given different input perturbations ($\underline{\mathbf{h}}_{i,j} \leq 0 \leq \overline{\mathbf{h}}_{i,j}$). In IBP training, inactive neurons have tighter bounds than active and unstable ones as shown in Figure 3 in Appendix B, and thus the optimization tends to push the neurons to be inactive. However, too many inactive neurons indicates that many neurons are essentially unused or dead, which will harm the model's capacity and block gradients as discussed by (Lu et al., 2019) on standard training.

### 3.3. The Proposed Method

To address the aforementioned issues, we propose our method in three parts: 1) We derive a new weight initialization for IBP training to stabilize the tightness of bounds at initialization; 2) We propose to fully add BNs to mitigate imbalanced ReLU and stabilize the variance of bounds; 3) We further propose regularizations to stabilize the tightness and the balance of ReLU neuron states during warmup.

#### 3.3.1. IBP INITIALIZATION

We propose *IBP initialization*. We independently initialize each element in $\mathbf{W}_i$ following a normal distribution $\mathcal{N}(0, \sigma_i^2)$, and we aim to choose a value for $\sigma_i$ such that the *difference gain* defined in Eq. (5) is exactly 1. When elements in $\mathbf{W}_i$ follow the normal distribution, we have $\mathbb{E}(|\mathbf{W}_i|) = \sqrt{2/\pi}\sigma_i$, and thereby we take $\sigma_i = \frac{\sqrt{2\pi}}{n_i}$, which makes the difference gain $\frac{n_i}{2}\mathbb{E}(|\mathbf{W}_i|)$ exactly 1.

#### 3.3.2. BATCH NORMALIZATION

For IBP, BN (Ioffe & Szegedy, 2015) can normalize the variance of bounds, and importantly, it can also improve the balance of ReLU activation states by shifting the center of upper and lower bounds to zero. But BN was partly or fully missed in prior certified training works (Gowal et al., 2018; Zhang et al., 2020; Xu et al., 2020). We will demonstrate the benefit of fully adding BN in the experiments. Shifting and scaling parameters of BN are computed from unperturbed data following Wong et al. (2018); Xu et al. (2020).

#### 3.3.3. WARMUP REGULARIZATION

We further add two regularizers to the warmup stage of IBP training, to explicitly stabilize the tightness of certified bounds and balance ReLU neuron states. It is principled and motivated by the identified issues.

**Bound tightness regularizer** We also expect to keep the mean value of $\Delta_i$ in the current batch, $\hat{\mathbb{E}}(\Delta_i)$, stable during the warmup. Here $\hat{\mathbb{E}}(\Delta_i)$ is empirically computed from a concrete batch and different from the expectation $\mathbb{E}(\Delta_i)$ in initialization. In our initialization, we aim to make $\mathbb{E}(\Delta_i) \approx \mathbb{E}(\Delta_{i-1})$ stable. Here we relax goal to making $\tau\hat{\mathbb{E}}(\Delta_i) \leq \hat{\mathbb{E}}(\Delta_0)$ with a configurable tolerance value $\tau$ ($0 < \tau \leq 1$), to balance the regularization power and the model capacity. We add the following regularization:

$$\mathcal{L}_{\text{tightness}} = \frac{1}{\tau m}\sum_{i=1}^{m}\text{ReLU}(\tau - \frac{\hat{\mathbb{E}}(\Delta_0)}{\hat{\mathbb{E}}(\Delta_i)}). \quad (6)$$

**ReLU activation states balancing regularizer** To balance ReLU activation states, we expect to balance the impact of active ReLU neurons and inactive neurons respectively. Here, we consider the center of the interval bound, $\mathbf{c}_i = (\underline{\mathbf{h}}_i + \overline{\mathbf{h}}_i)/2$, and we model the impact as the contribution of each type of neurons to the mean and variance of the whole layer, i.e., $\hat{\mathbb{E}}(\mathbf{c}_i)$ and $\text{Var}(\mathbf{c}_i)$ respectively. We use $\alpha_i$ to denote the ratio between the contribution of the active neurons and inactive neurons respectively to $\hat{\mathbb{E}}(\mathbf{c}_i)$, and similarly we use $\beta_i$ for $\text{Var}(\mathbf{c}_i)$. We compute: $\alpha_i = \frac{\sum_j \mathbb{I}(\underline{\mathbf{h}}_{i,j}>0)\mathbf{c}_{i,j}}{-\sum_j \mathbb{I}(\overline{\mathbf{h}}_{i,j}<0)\mathbf{c}_{i,j}}$, $\beta_i = \frac{\sum_j \mathbb{I}(\underline{\mathbf{h}}_{i,j}>0)(\mathbf{c}_{i,j}-\hat{\mathbb{E}}(\mathbf{c}_i))^2}{\sum_j \mathbb{I}(\overline{\mathbf{h}}_{i,j}<0)(\mathbf{c}_{i,j}-\hat{\mathbb{E}}(\mathbf{c}_i))^2}$, where $\mathbf{h}_{i,j}, \underline{\mathbf{h}}_{i,j}, \overline{\mathbf{h}}_{i,j}$ stand for the value and bounds of each neuron in layer $i$, and in general $\alpha_i, \beta_i > 0$ unless in the training start. With the same aforementioned tolerance $\tau$, we expect to make $\tau \leq \alpha_i, \beta_i \leq 1/\tau$, which is equivalent to making $\min(\alpha_i, 1/\alpha_i) \geq \tau$, $\min(\beta_i, 1/\beta_i) \geq \tau$. Thereby we design the following regularization term:

$$\mathcal{L}_{\text{relu}} = \frac{1}{\tau m}\sum_{i=1}^{m}(\text{ReLU}(\tau - \min(\alpha_i, \frac{1}{\alpha_i}))$$
$$+\text{ReLU}(\tau - \min(\beta_i, \frac{1}{\beta_i}))). \quad (7)$$

*Table 1.* Standard and verified error rates (%) of models trained with different methods respectively on CIFAR-10 ($\epsilon_{\text{target}} = 8/255$). Schedule is represented as the total number of epochs and the number of epochs in each phase (in the parentheses), $\epsilon = 0$, increasing $\epsilon \in (0, \epsilon_{\text{target}})$ and final $\epsilon = \epsilon_{\text{target}}$ respectively. We report the mean and standard deviation of the results on 5 repeats for CNN-7 and 3 repeats for Wide-ResNet and ResNeXt respectively. We also report the result of our best run in "Ours (best)", since main results in prior works did not have repeats, and we include literature results for reference. Literatures with the "†" mark are concurrent preprint works.

| Dataset | Schedule (epochs) | Method | CNN-7 | | Wide-ResNet | | ResNeXt | |
|---|---|---|---|---|---|---|---|---|
| | | | Standard | Verified | Standard | Verified | Standard | Verified |
| CIFAR-10 | 70 (1+20+49) | Vanilla IBP | $58.72 \pm 0.27$ | $69.88 \pm 0.10$ | $58.85 \pm 0.22$ | $69.77 \pm 0.32$ | $60.10 \pm 0.27$ | $71.19 \pm 0.21$ |
| | | CROWN-IBP | $63.19 \pm 0.36$ | $71.29 \pm 0.19$ | $62.76 \pm 0.23$ | $71.82 \pm 0.30$ | $64.75 \pm 0.50$ | $72.50 \pm 0.20$ |
| | | Ours | $\mathbf{56.64 \pm 0.48}$ | $\mathbf{68.81 \pm 0.24}$ | $\mathbf{56.74 \pm 0.40}$ | $\mathbf{68.71 \pm 0.29}$ | $\mathbf{59.33 \pm 0.86}$ | $\mathbf{70.62 \pm 0.59}$ |
| | 160 (1+80+79) | Vanilla IBP | $53.80 \pm 0.71$ | $67.01 \pm 0.29$ | $54.31 \pm 0.46$ | $67.45 \pm 0.21$ | $55.23 \pm 0.12$ | $68.28 \pm 0.15$ |
| | | CROWN-IBP | $58.76 \pm 0.76$ | $69.67 \pm 0.38$ | $60.39 \pm 0.33$ | $70.07 \pm 0.42$ | $61.08 \pm 0.35$ | $71.26 \pm 0.11$ |
| | | Ours | $\mathbf{51.72 \pm 0.40}$ | $\mathbf{65.58 \pm 0.32}$ | $\mathbf{51.95 \pm 0.27}$ | $\mathbf{65.91 \pm 0.14}$ | $\mathbf{53.68 \pm 0.33}$ | $\mathbf{66.91 \pm 0.40}$ |
| | | Ours (best) | **51.06** | **65.03** | 51.63 | 65.72 | 53.38 | 66.41 |
| | Literature results | | Warmup | | Total (epochs) | | Standard | Verified |
| | Gowal et al. (2018) | | (5K+50K) steps | | 3,200 | | 50.51 | 68.44 |
| | Zhang et al. (2020) | | (320 + 1600) epochs | | 3,200 | | 54.02 | 66.94 |
| | Balunovic & Vechev (2020) | | N/A [a] | | 800 | | 48.3 | 72.5 |
| | Xu et al. (2020) | | (100 + 800) epochs | | 2,000 | | 53.71 | 66.62 |
| | †IBP+ParamRamp (Lyu et al., 2021) | | (320 + 1600) epochs | | 3,200 | | 55.28 | 67.09 |
| | †CROWN-IBP+ParamRamp (Lyu et al., 2021) | | (320 + 1600) epochs | | 3,200 | | 51.94 | 65.08 |
| | †$\ell_\infty$-dist net (other architecture) (Zhang et al., 2021) [b] | | N/A [b] | | 800 | | 48.32 | 64.90 |

[a] Balunovic & Vechev (2020) used a different training scheme and train the network layer by layer.
[b] Concurrent Zhang et al. (2021) use a very different model architecture with $\ell_\infty$ distance neurons rather than traditional DNNs, but still need a long schedule on both $\epsilon$ and $\ell_p$ norm where $p$ is gradually increased until $\infty$.

**Training Objectives** The overall training objective is $\mathcal{L} = \overline{L}(\mathbf{x}, y, \epsilon) + \lambda(\mathcal{L}_{\text{tightness}} + \mathcal{L}_{\text{relu}})$, where $\overline{L}(\mathbf{x}, y, \epsilon)$ is an original IBP loss, and $\lambda$ is for balancing the original loss and regularization. During warmup, we gradually decrease $\lambda$ from $\lambda_0$ to 0 as $\epsilon$ grows to $\epsilon_{\text{target}}$, where $\lambda = \lambda_0(1 - \epsilon/\epsilon_{\text{target}})$.

*Table 2.* Comparison of estimated time cost (seconds), for CNN-7 on CIFAR-10. We also include the total training cost of literature works using long schedules, where literatures with the "†" mark are concurrent works.

| Method | Epochs | Total |
|---|---|---|
| IBP (Gowal et al., 2018) | 3200 | $40496 \times 4$ [a] |
| CROWN-IBP (w/o loss fusion) (Zhang et al., 2020) | 3200 | $91288 \times 4$ [a] |
| CROWN-IBP (Xu et al., 2020) | 2000 | $52362 \times 4$ [a] |
| †IBP+ParamRamp (Lyu et al., 2021) | 3200 | $40496 \times 4 \times 1.09$ [b] |
| †CROWN-IBP+ParamRamp (Lyu et al., 2021) | 3200 | $91288 \times 4 \times 1.51$ [b] |
| Vanilla IBP (verified error 67.01±0.29) | 160 | 8747.9 |
| CROWN-IBP (verified error 69.67±0.38) | 160 | 10641.3 |
| Ours (verified error 65.58±0.32) | 160 | 9512.3 |

[a]They used 4 GPUs; [b]Overhead factor by Lyu et al. (2021).

## 4. Experimental Results

### 4.1. Settings

We adopt two datasets, CIFAR-10 (Krizhevsky et al., 2009) and TinyImageNet (Le & Yang, 2015). We mainly compare our improved IBP training with two baselines. **Vanilla IBP** (Gowal et al., 2018) with existing initialization and no warmup regularizer; **CROWN-IBP** (Zhang et al., 2020; Xu et al., 2020) that combines IBP bounds and linear relaxation bounds with weight $\frac{\epsilon}{\epsilon_{\text{target}}}$ and $(1 - \frac{\epsilon}{\epsilon_{\text{target}}})$ respectively during training. We report more details in Appendix C.

### 4.2. Certified Robust Training with Short Warmup

We use relatively short warmup schedules to demonstrate for fast training, and we show our results in Table 1 for CIFAR-

10, and Appendix B.2 for TinyImageNet. Compared to Vanilla IBP and CROWN-IBP, our improved IBP training consistently achieves lower errors under same schedules respectively, where BN is added to the models for all these three training methods. We achieve verified error $65.03\%$ on CIFAR-10 $\epsilon = 8/255$, and $82.36\%$ on TinyImageNet $\epsilon_{\text{target}} = 1/255$, which significantly outperform literature SOTA (Gowal et al., 2018; Xu et al., 2020). Compared to concurrent preprint works (Lyu et al., 2021; Zhang et al., 2021) with different improvement techniques, we have comparable verified errors, but much shorter training schedule. We tried the code of concurrent Zhang et al. (2021), on CIFAR-10 using 160 total epochs by reducing their training schedule proportionally, their verified error is 68.44% which is much higher than ours.

### 4.3. Comparison on Training Cost

We compare the training cost of different methods, using a single Nvidia RTX 2080 Ti GPU. We show results of CNN-7 for CIFAR-10 in Table 2, and others in Appendix B. We only have a small overhead compared to Vanilla IBP and the cost is still around lower than CROWN-IBP, while we achieve lower verified errors than the baselines (see Table 1). And importantly, compared to literatures using long training schedules, we significantly reduce total training time.

## 5. Conclusion

In this paper, we propose to improve IBP training and reduce the length of warmup schedule for fast training. We are able to achieve better verified errors using much shorter training schedules compared to literatures under the same convolution-based network architecture.

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

*Table 3.* List of several weight initialization methods and their *difference gain*. We show each difference gain in both closed form, and also empirical values when $n_i \in \{27, 576, 1152, 32768\}$ for a 7-layer CNN model in our experiments. The concrete values are obtained by computing the mean of 100 random trials respectively. For orthogonal initialization, obtaining a closed form of difference gain is non-trivial so we omit its closed-form result, but it has large difference gains under empirical measurements.

| Method | Adopted by | Difference Gain | | | | |
|---|---|---|---|---|---|---|
| | | Closed form | $n_i = 27$ | $n_i = 576$ | $n_i = 1152$ | $n_i = 32768$ |
| Xavier (uniform) (Glorot & Bengio, 2010) | Zhang et al. (2020); Xu et al. (2020) | $\frac{1}{4}\sqrt{n_i}$ | 1.30 | 6.00 | 8.48 | 45.25 |
| Orthogonal (Saxe et al., 2013) | Gowal et al. (2018) | - | 2.09 | 9.58 | 13.54 | 72.22 |
| Kaiming (uniform) (He et al., 2015b) | - | $\frac{\sqrt{3}}{4}\sqrt{n_i}$ | 3.20 | 14.70 | 20.77 | 110.85 |
| IBP Initialization | This work | 1 | 1.01 | 1.00 | 1.00 | 1.00 |

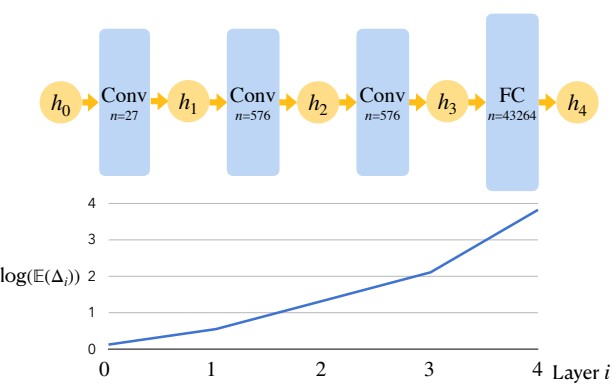

*Figure 2.* We show a simple untrained CNN (the classification layer is omitted) with Xavier initialization. We evaluate the mean of each layer's $\Delta_i$ as an estimation of $\mathbb{E}(\Delta_i)$ and plot $\log \mathbb{E}(\Delta_i)$. Interval bounds explode in deeper layers.

## A. Supplementary Illustrations for Motivation and Methodology

### A.1. List of Initialization Methods in Prior Works

In Table 3, we list several weight initialization methods and their corresponding difference gain. Prior weight initialization methods lead to large difference gain values especially when $n_i$ is larger, which indicates exploded certified bounds at initialization. In contrast, our initialization yields a constant difference gain of 1 regardless of $n_i$.

### A.2. Illustration of Exploded Bounds using Existing Initialization

In Figure 2, we illustrate that certified bounds can explode for a model initialized using prior Xavier (Glorot & Bengio, 2010) initialization.

### A.3. Illustration of IBP Relaxations for Different Neuron States

In Figure 3, we illustrate IBP relaxations for ReLU neurons with the three different states respectively. Inactive neurons have no relaxation error compared with the other two kinds

of neurons, and thus IBP training tends to prefer inactive neurons more to tighten certified bounds, compared to the other two ReLU neuron states. This leads to an imbalance in ReLU neuron states for vanilla IBP on models without BN. In this paper, we identify the benefit of fully adding BN layers to mitigate the imbalance, because BN normalizes pre-activation values. We also add a regularization to further encourage ReLU balance.

## B. Additional Experiments

### B.1. Ablation Study and Discussions

In this section, we empirically verify whether each part of our modification contribute to the improvement and whether they behave as we expect. We first conduct an ablation study and we also plot the curve of the regularization terms to reflect the bound tightness and ReLU balance in different settings.

In the ablation study, we use CIFAR-10 with the currently best CNN-7 model under the "$1+20$" and "$1+80$" warmup schedules as used in Table 1. We start from a vanilla setting, and we add BN, IBP initialization, and the warmup regularizers to the model or training. We report the results in Table 4. The first three rows show that fully adding BN improves the training when vanilla IBP is used, and it is important to add BN for the fully-connected layer, which was missed in prior works. Based on the improved model structure, adding both IBP initialization and warmup regularization further improves the performance, and removing either of these parts leads to a degraded performance.

We notice that adding IBP initialization alone may not necessarily bring improvement to the verified error. A factor is that IBP initialization can reduce the variance of the outputs, as discussed in Appendix D.2, and it may harm the training during the early warmup when $\epsilon$ is small and certified training is close to standard training. Also, the effect of initialization can be weakened during the warmup when $\epsilon$ is much smaller than $\epsilon_{\text{target}}$. But when we combine it with the regularizers, the regularization can continue to tighten the bounds, and the IBP initialization can benefit the optimization for the tightness regularizer. Nevertheless, IBP

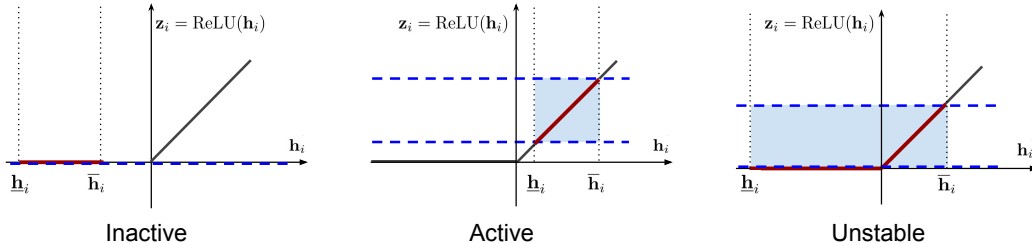

*Figure 3.* Three activation states of ReLU neurons determined by pre-activation lower and upper bounds and their corresponding IBP relaxations. The relaxed areas are shown in light blue.

*Table 4.* Ablation study results. We use the CNN-7 model on CIFAR-10. "BN-Conv" stands for BN layers after each convolutional layer, and "BN-FC" stands for BN layers after the hidden fully-connected layer. "✓" means that the component is enabled, and "×" means that the component is disabled. We repeat each setting for 5 times and report the mean and standard deviation.

| BN-Conv | BN-FC | IBP Initialization | $\mathcal{L}_{\text{tightness}}$ | $\mathcal{L}_{\text{relu}}$ | 70 (1+20+49) | | 160 (1+80+79) | |
| --- | --- | --- | --- | --- | --- | --- | --- | --- |
| | | | | | Standard (%) | Verified (%) | Standard (%) | Verified (%) |
| × | × | × | × | × | 59.33±0.70 | 70.18±0.18 | 57.08±0.29 | 69.43±0.28 |
| ✓ | × | × | × | × | 61.95±0.80 | 71.12±0.42 | 57.21±0.65 | 69.21±0.30 |
| ✓ | ✓ | × | × | × | 58.72±0.27 | 69.88±0.10 | 53.80±0.71 | 67.01±0.29 |
| ✓ | ✓ | ✓ | × | × | 58.93±0.29 | 69.60±0.35 | 54.59±0.64 | 67.63±0.34 |
| ✓ | ✓ | ✓ | ✓ | × | 56.76±0.38 | 68.96±0.49 | 53.08±0.26 | 66.74±0.20 |
| ✓ | ✓ | ✓ | × | ✓ | 58.49±0.42 | 69.38±0.23 | 53.29±0.76 | 66.46±0.44 |
| ✓ | ✓ | × | ✓ | ✓ | 58.79±0.40 | 69.29±0.28 | 52.45±0.34 | 66.34±0.38 |
| ✓ | ✓ | ✓ | ✓ | ✓ | **56.64±0.48** | **68.81±0.24** | **51.72±0.40** | **65.58±0.32** |

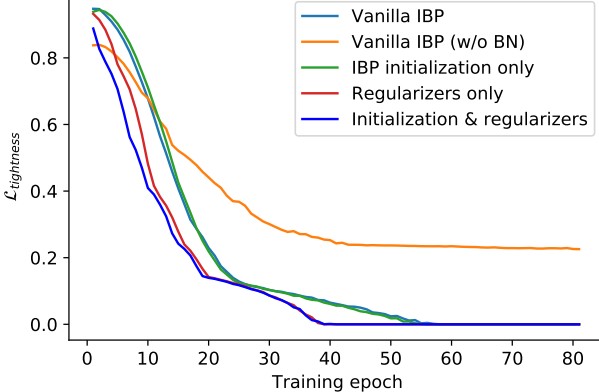

*Figure 4.* $\mathcal{L}_{\text{tightness}}$ during warmup. $\mathcal{L}_{\text{tightness}}$ is optimized only for "regularizers only" and "initialization & regularizers" setting, and BN is fully added except for "Vanilla IBP (w/o BN)".

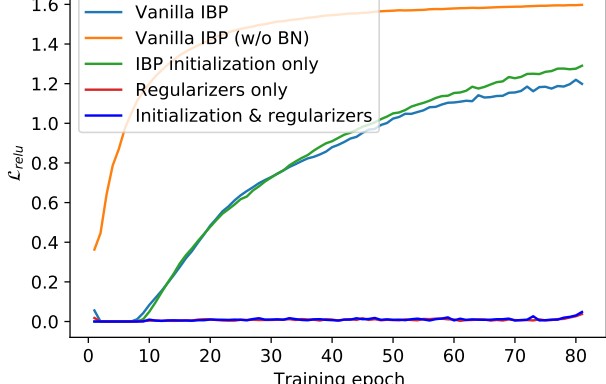

*Figure 5.* $\mathcal{L}_{\text{relu}}$ during warmup, with same settings as in Figure 4.

initialization is more beneficial for deep models where the exploded bound issue is more severe. In Figure 6, we show that for a ResNeXt on TinyImageNet, IBP initialization is helpful for reaching lower verified errors especially at early epochs.

In Figure 4, we plot the $\mathcal{L}_{\text{tightness}}$ during training for different settings. Note that for the settings without regularizers, we only plot the loss terms but not optimize them during training. By using the regularization in training, $\mathcal{L}_{\text{tightness}}$ de-

scends faster, and further adding the IBP initialization leads to even faster descent during the early epochs. In Figure 5, we show that the $\mathcal{L}_{\text{relu}}$ term is indeed under control with our regularizer added in training, which indicates the ReLU activation states is more balanced during training, while $\mathcal{L}_{\text{relu}}$ could gradually grow larger when the regularization is not added in training. Notably, when BN is removed and the regularization term is not optimized (Vanilla IBP (w/o BN)), $\mathcal{L}_{\text{relu}}$ becomes extremely large in later epochs, and $\mathcal{L}_{\text{tightness}}$ is also large in the end, which suggests the training is hampered.

*Table 5.* Standard and verified error rates (%) of models trained on TinyImageNet ($\epsilon_t = 1/255$). The best result in literature (Xu et al., 2020) is standard error 72.18% and verified error 84.14% using 800 epochs.

| Model | Schedule (epochs) | Vanilla IBP w/o BN | | Vanilla IBP | | CROWN-IBP | | Ours | |
|---|---|---|---|---|---|---|---|---|---|
| | | Standard | Verified | Standard | Verified | Standard | Verified | Standard | Verified |
| CNN-7 | 80 (1+10+69) | 80.28 | 86.59 | 75.50 | 82.92 | 76.00 | 82.81 | 75.20 | **82.45** |
| | 80 (1+20+59) | 79.35 | 86.06 | 74.68 | 82.84 | 76.27 | 83.35 | 74.29 | **82.36** |
| Wide-Resnet [a] | 80 (1+10+69) | 79.26 | 85.40 | 75.89 | 83.00 | 75.85 | 83.65 | 74.90 | **82.49** |
| | 80 (1+20+59) | 78.45 | 85.19 | 75.65 | 83.17 | 75.95 | 83.08 | 74.59 | **82.75** |
| ResNext | 80(1+10+69) | 83.27 | 88.14 | 82.39 | 87.15 | 85.47 | 89.11 | 80.20 | **85.77** |
| | 80 (1+20+59) | 82.04 | 87.88 | 81.72 | 87.10 | 80.81 | 86.43 | 78.91 | **85.78** |

[a] The Wide-ResNet model used here is 5 times smaller than the one used in (Xu et al., 2020) to save cost.

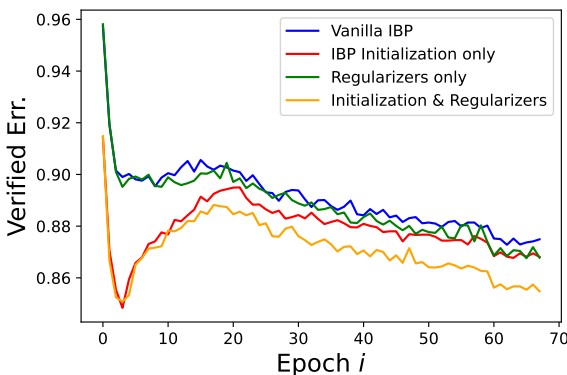

*Figure 6.* Curve of training verified error of a ResNeXt on TinyImageNet. Note that the verified errors can increase during the warmup as $\epsilon$ increases.

## B.2. Results on Tiny-ImageNet

In Table B.2, we report standard and verified error results on TinyImageNet. Our improved method consistently outperforms the baselines.

## B.3. Computational Cost for All Datasets and Models

In this section, we report full results on the computational cost comparison for all datasets and models. We measure the per-epoch time during three $\epsilon$ phases, and we then estimate the total training time according to the schedule. We show the results in Table 6. In addition to the time cost comparison on CNN-7 on CIFAR shown in Section 4.3, we report computation cost results for all the datasets and models in Table 6. For $\epsilon = 0$, Vanilla IBP and CROWN-IBP use regular training while we compute IBP bounds for regularization and have a small overhead, but this phase is extremely short (no more than 1 epoch here) and thus negligible. For $0 < \epsilon < \epsilon_{\text{target}}$, our method has a smaller overhead on regularizers compared to Vanilla IBP, while CROWN-IBP using linear relaxation can be more costly. In $\epsilon = \epsilon_{\text{target}}$, all the three methods use the same pure IBP.

Under same training schedules, results show that our pro-

posed method has a small overhead over vanilla IBP, and the cost is still lower than that of CROWN-IBP. Meanwhile, our method is able to achieve lower verified errors compared to the two baselines (Table 1 and Table 5). More importantly, we are able to use much shorter training schedules to achieve SOTA results compared to prior literatures, to enable fast certified robust training, shown in Table 2.

## B.4. Other Perturbation Radii

In Table 7, we present results using perturbation radii other than those used in our main experiments. Here we consider $\epsilon_{\text{target}} \in \{0.1, 0.3\}$ for MNIST, and $\epsilon_{\text{target}} \in \{\frac{2}{255}, \frac{16}{255}\}$ for CIFAR-10. In particular, on MNIST models are trained with target perturbation radii $\epsilon_{\text{train}}$ larger than used for testing $\epsilon_{\text{target}}$ to mitigate overfitting – we use $\epsilon_{\text{train}} = 0.2$ when $\epsilon_{\text{target}} = 0.1$ and $\epsilon_{\text{train}} = 0.4$ when $\epsilon_{\text{target}} = 0.3$ following Zhang et al. (2020). We use the CNN-7 model in this experiment. Results show that improvements over Vanilla IBP and CROWN-IBP are consistent as in Table 1. Overall, the experimental results demonstrate that our proposed method is effective on settings with different perturbation radii, compared to vanilla IBP and CROWN-IBP.

## B.5. ReLU Imbalance with Shorter Warmup Length

In Figure 1, we show two 7-layer CNN models with different warmup length respectively, and the model tends to have more inactive neurons and thus more severe imbalance in ReLU neuron states for shorter warmup length, as previously mentioned in Section 3.2.

## C. Experiment Details

**Implementation** Our implementation is based on the `auto_LiRPA` (Xu et al., 2020) library[1] for robustness verification and certified training on general computational graphs. Baselines including Vanilla IBP and CROWN-IBP with loss fusion are inherently supported by the library. In `auto_LiRPA`, Xavier initialization (Glorot & Bengio,

---

[1] https://github.com/KaidiXu/auto_LiRPA

*Table 6.* Comparison of estimated time cost (seconds) on all the datasets and models. We report the per-epoch time during training phases with different $\epsilon$ ranges, and we report the total time when the $0 + 20 + 50$ schedule is used for MNIST, the $1 + 80 + 79$ schedule is used for CIFAR-10, and the $1 + 10 + 69$ schedule for TinyImageNet respectively. "-" in the table means that there is no $\epsilon = 0$ warmup stage for MNIST following Zhang et al. (2020). Note that on each dataset, for phases of same or different methods that are supposed to be equivalent in algorithm implementation, we make them share the same time estimation result respectively.

| Dataset | Model | Method | Per-epoch for $\epsilon$ | | | Total |
| | | | 0 | $(0, \epsilon_{target})$ | $\epsilon_{target}$ | |
|---|---|---|---|---|---|---|
| MNIST | CNN-7 | Vanilla IBP | - | 27.9 | 27.9 | 1955.1 |
| | | CROWN-IBP | - | 49.6 | 27.9 | 2387.5 |
| | | Ours | - | 37.0 | 27.9 | 2135.8 |
| | Wide-ResNet | Vanilla IBP | - | 81.0 | 81.0 | 5668.3 |
| | | CROWN-IBP | - | 142.1 | 81.0 | 6890.2 |
| | | Ours | - | 99.0 | 81.0 | 6029.3 |
| | ResNeXt | Vanilla IBP | - | 73.2 | 73.2 | 5127.2 |
| | | CROWN-IBP | - | 147.7 | 73.2 | 6616.9 |
| | | Ours | - | 104.4 | 73.2 | 5750.7 |
| CIFAR-10 | CNN-7 | Vanilla IBP | 30.0 | 54.8 | 54.8 | 8747.9 |
| | | CROWN-IBP | 30.0 | 78.5 | 54.8 | 10641.3 |
| | | Ours | 64.0 | 64.0 | 54.8 | 9512.3 |
| | Wide-ResNet | Vanilla IBP | 43.7 | 114.7 | 114.7 | 18358.4 |
| | | CROWN-IBP | 43.7 | 170.7 | 114.7 | 22764.9 |
| | | Ours | 134.7 | 134.7 | 114.7 | 19976.0 |
| | ResNeXt | Vanilla IBP | 38.7 | 102.7 | 102.7 | 16432.0 |
| | | CROWN-IBP | 38.7 | 183.3 | 102.7 | 22813.6 |
| | | Ours | 129.6 | 129.6 | 102.7 | 18611.7 |
| TinyImageNet | CNN-7 | Vanilla IBP | 282.2 | 431.4 | 431.4 | 34362.0 |
| | | CROWN-IBP | 282.2 | 663.8 | 431.4 | 36686.5 |
| | | Ours | 500.4 | 500.4 | 431.4 | 35270.3 |
| | Wide-ResNet | Vanilla IBP | 270.2 | 399.8 | 399.8 | 31861.6 |
| | | CROWN-IBP | 270.2 | 592.1 | 399.8 | 33789.3 |
| | | Ours | 464.6 | 464.6 | 399.8 | 32703.0 |
| | ResNeXt | Vanilla IBP | 197.2 | 430.5 | 430.5 | 34206.7 |
| | | CROWN-IBP | 197.2 | 883.1 | 430.5 | 38735.1 |
| | | Ours | 626.3 | 626.3 | 430.5 | 36595.8 |

2010) is used by default, which is also the default initialization in PyTorch for regular DNN training. We find that orthogonal initialization (Saxe et al., 2013) originally used by (Gowal et al., 2018) does not seem to improve the performance over Xavier initialization. We add to implement our IBP initialization and warmup with regularizers for fast certified robust training.

**Datasets** For MNIST and CIFAR-10, we load the datasets using `torchvision.datasets`[2] and use the original data splits. On CIFAR-10, we use random horizontal flips and random cropping for data augmentation, and also normalize input images, following Zhang et al. (2020); Xu et al. (2020). For TinyImageNet, we download the dataset from Stanford CS231n course website[3]. Similar to CIFAR-10, we also use data augmentation and normalize input images for TinyImageNet. Unlike Xu et al. (2020) which cropped the $64 \times 64$ original images into $56 \times 56$ and used a central

---

[2]https://pytorch.org/vision/0.8/datasets.html
[3]http://cs231n.stanford.edu/TinyImageNet-200.zip

$56 \times 56$ cropping for test images, we pad the cropped training images back to $64 \times 64$ so that we do not need to crop test images. We use the validation set for testing since test images are unlabelled, following Xu et al. (2020).

**Models** We use three model architectures in the experiments: a 7-layer feedforward convolutional network (CNN-7), Wide-ResNet (Zagoruyko & Komodakis, 2016) and ResNeXt (Xie et al., 2017). All the models have a hidden fully-connected layer with 512 neurons prior to the classification layer. For CNN-7, there are five convolutional layers with $64, 64, 128, 128, 128$ filters respectively. For Wide-ResNet, there are 3 wide basic blocks, with a widen factor of 8 for MNIST and CIFAR-10 and 10 for TinyImageNet. For ResNeXt, we use $1, 1, 1$ blocks for MNIST and CIFAR-10, and $2, 2, 2$ blocks for TinyImageNet; the cardinality is set to 2, and the bottleneck width is set to 32 for MNIST and CIFAR-10 and 8 for TinyImageNet. For all the models, ReLU is used as the activation. These models were similarly adopted in Xu et al. (2020). But we fully add BNs after each convolutional layer and fully-connected layer, while some of these BNs were missed in Xu et al.

*Table 7.* The standard errors (%) and verified errors (%) of a CNN-7 model trained with different methods on other perturbation radii not included in the main results.

| Dataset | Warmup | $\epsilon_{\text{target}}$ | $\epsilon_{\text{train}}$ | Vanilla IBP | | CROWN-IBP | | Ours | |
|---|---|---|---|---|---|---|---|---|---|
| | | | | Standard | Verified | Standard | Verified | Standard | Verified |
| MNIST | 0+20 | 0.1 | 0.2 | 1.12 | 2.17 | 1.07 | 2.17 | 1.16 | **2.05** |
| | | 0.3 | 0.4 | 2.74 | 7.61 | 2.88 | 7.55 | 2.33 | **6.90** |
| CIFAR-10 | 1+80 | 2/255 | | 33.65 | 48.75 | 34.09 | 48.28 | 33.16 | **47.15** |
| | | 16/255 | | 64.52 | 76.36 | 71.75 | 79.43 | 63.35 | **75.52** |

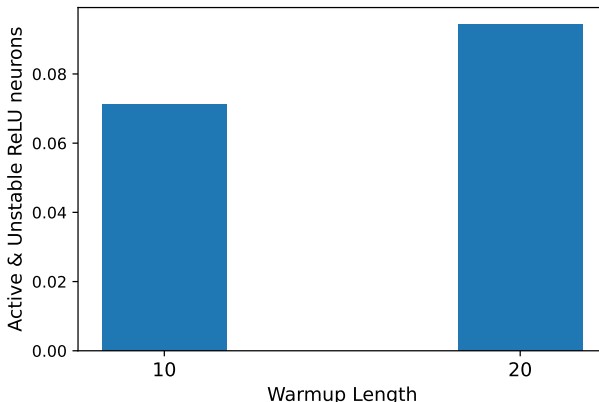

*Figure 7.* Ratio of active and unstable neurons in two 7-layer CNN models trained with Vanilla IBP using different warmup lengths respectively.

(2020). For example, the CNN-7 model in Xu et al. (2020) had BN for convolutional layers but not the fully-connected layer. Besides, we remove the average pooling layer in Wide-ResNet as we find it harms the performance of all the considered training methods, and this modification makes the Wide-ResNet align better with the CNN-7 model, which does not have average pooling either but achieves best results compared to other models (Table 1 and Table 5).

**Training** During certified training, models are trained with Adam (Kingma & Ba, 2014) optimizer with an initial learning rate of $5 \times 10^{-4}$, and there are two milestones where the learning rate decays by 0.2. We determine the milestones for learning rate decay according to the training schedule and the total number of epochs, as shown in Table 8. Gradient clipping threshold is set to 10.0. We train the models using a batch size of 256 on MNIST, and 128 on CIFAR-10 and TinyImageNet. The tolerance value $\tau$ in our warmup regularization is fixed to 0.5. For Vanilla IBP and IBP with our initialization and regularizers, we train the models on a single Nvidia GeForce GTX 1080 Ti or Nvidia GeForce RTX 2080 Ti GPU. For CROWN-IBP, we train the models on two GPUs for efficiency, while in time estimation we still use one single GPU for fair comparison. The number of training and evaluation runs is 1 for each experiment result respectively. In the evaluation, the major

metric is *verified error*, which stands for the rate of test examples such that the model cannot certifiably make correct predictions given the $\ell_\infty$ perturbation radius. For reference, we also report *standard error*, which is the standard error rate where no perturbation is considered.

*Table 8.* Milestones for learning rate decay when different total number of epochs are used. "Decay-1" and "Decay-2" denote the two milestones respectively when the learning rate decays by a factor of 0.2.

| Dataset | Total epochs | Decay-1 | Decay-2 |
|---|---|---|---|
| MNIST | 50 | 40 | 45 |
| | 70 | 50 | 60 |
| CIFAR-10 | 70 | 50 | 60 |
| | 160 | 120 | 140 |
| TinyImageNet | 80 | 60 | 70 |

**Warmup scheduling** During the warmup stage, after training with $\epsilon = 0$ for a number of epochs, the perturbation radius $\epsilon$ is gradually increased from 0 until the target perturbation radius $\epsilon_{\text{target}}$, during the $0 < \epsilon < \epsilon_{\text{target}}$ phase. Specifically, during the first 25% epochs of the $\epsilon$ increasing stage, $\epsilon$ is increased exponentially, and after that $\epsilon$ is increased linearly. In this way, $\epsilon$ remains relatively small and increases relatively slowly during the beginning, to stabilize training. We use the `SmooothedScheduler` in the `auto_LiRPA` as the scheduler for $\epsilon$ similarly adopted by Xu et al. (2020). On CIFAR-10, unlike some prior works which made the perturbation radii used for training 1.1 times of those for testing respectively (Gowal et al., 2018; Zhang et al., 2020), we find this setting makes little improvement over using same perturbation radii for both training and testing in our experiments as also mentioned in Lee et al. (2021), and thus we directly adopt the later setting for simplicity.

## D. Proofs

### D.1. Proof of Eq. (4)

In this section, we provide a proof for Eq. (4):

$$\mathbb{E}(\delta_i) = \mathbb{E}(\text{ReLU}(\overline{\mathbf{h}}_i)) - \text{ReLU}(\underline{\mathbf{h}}_i)) = \frac{1}{2}\mathbb{E}(\Delta_i), \quad (8)$$

where $\Delta_i = \overline{\mathbf{h}}_i - \underline{\mathbf{h}}_i$, and $\delta_i = \overline{\mathbf{z}}_i - \underline{\mathbf{z}}_i$.

*Proof.* We first have

$$\mathbb{E}(\delta_i) = \mathbb{E}(\text{ReLU}(\overline{\mathbf{h}}_i) - \text{ReLU}(\underline{\mathbf{h}}_i))$$

$$= \mathbb{E}(\text{ReLU}(\mathbf{c}_i + \frac{\Delta_i}{2}) - \text{ReLU}(\mathbf{c}_i - \frac{\Delta_i}{2})) \quad (9)$$

$$= \mathbb{E}(\text{ReLU}(\mathbf{c}_i + \frac{\Delta_i}{2})) - \mathbb{E}(\text{ReLU}(\mathbf{c}_i - \frac{\Delta_i}{2})).$$

Note that $\mathbf{c}_i = \frac{1}{2}\mathbf{W}_i(\underline{\mathbf{z}}_i + \overline{\mathbf{z}}_i)$ and $\Delta_i = |\mathbf{W}_i|\delta_i$, and thus $p(-\mathbf{c}_i \mid |\mathbf{W}_i|) = p(\mathbf{c}_i \mid |\mathbf{W}_i|)$ and $p(-\mathbf{c}_i|\Delta_i) = p(\mathbf{c}_i|\Delta_i)$, where we use $p(\cdot)$ to denote the probability density function (PDF). Thereby,

$$\mathbb{E}(\text{ReLU}(\mathbf{c}_i + \frac{\Delta_i}{2}))$$
$$= \int_0^\infty \int_{-\frac{\Delta_i}{2}}^\infty (\mathbf{c}_i + \frac{\Delta_i}{2})p(\mathbf{c}_i|\Delta_i)p(\Delta_i)d\mathbf{c}_i d\Delta_i,$$
$$\mathbb{E}(\text{ReLU}(\mathbf{c}_i - \frac{\Delta_i}{2})) \quad (10)$$
$$= \int_0^\infty \int_{\frac{\Delta_i}{2}}^\infty (\mathbf{c}_i - \frac{\Delta_i}{2})p(\mathbf{c}_i|\Delta_i)p(\Delta_i)d\mathbf{c}_i d\Delta_i.$$

And thus

$$\mathbb{E}(\text{ReLU}(\mathbf{c}_i + \frac{\Delta_i}{2})) - \mathbb{E}(\text{ReLU}(\mathbf{c}_i - \frac{\Delta_i}{2}))$$
$$= \int_0^\infty (\int_{\frac{\Delta_i}{2}}^\infty \Delta_i + \int_{-\frac{\Delta_i}{2}}^{\frac{\Delta_i}{2}} (\mathbf{c}_i + \frac{\Delta_i}{2}))p(\mathbf{c}_i|\Delta_i)p(\Delta_i)d\mathbf{c}_i d\Delta_i$$
$$= \int_0^\infty \int_{-\infty}^\infty \frac{\Delta_i}{2}p(\mathbf{c}_i|\Delta_i)p(\Delta_i)d\mathbf{c}_i d\Delta_i$$
$$= \frac{1}{2}\mathbb{E}(\Delta_i).$$
$$(11)$$
$$\square$$

**D.2. Proof on the Bounds of $\text{Var}(\underline{\mathbf{h}}_i)$ and $\text{Var}(\overline{\mathbf{h}}_i)$**

In this section, we show that $\text{Var}(\underline{\mathbf{h}}_i)$ and $\text{Var}(\overline{\mathbf{h}}_i)$ will not explode or vanish at initialization, so that the magnitude of forward signals will not vanish or explode when we use IBP initialization which focuses on stabilizing the tightness of certified bounds.

We can derive that

$$\text{Var}(\overline{\mathbf{h}}_i) = \text{Var}(\mathbf{W}_{i,+}\overline{\mathbf{z}}_{i-1} + \mathbf{W}_{i,-}\underline{\mathbf{z}}_{i-1})$$
$$= \text{Var}([\mathbf{W}_{i,+}\overline{\mathbf{z}}_{i-1} + \mathbf{W}_{i,-}\underline{\mathbf{z}}_{i-1}]_j) \ (0 \leq j \leq r_i)$$
$$= \text{Var}(\sum_{k=1}^{n_i} ([\mathbf{W}_i]_{j,k}[\overline{\mathbf{z}}_{i-1}]_k \cdot \mathbb{I}([\mathbf{W}_i]_{j,k} > 0)))$$
$$+ \sum_{k=1}^{n_i} ([\mathbf{W}_i]_{j,k}[\underline{\mathbf{z}}_{i-1}]_k \cdot \mathbb{I}([\mathbf{W}_i]_{j,k} \leq 0)))).$$

Since $\mathbf{W}_i$ is initialized with mean 0, the numbers of negative elements and positive elements are approximately equal, and

thus

$$\text{Var}(\overline{\mathbf{h}}_i) \approx \frac{n_i}{2}\text{Var}(\mathbf{W}_{i,+}\overline{\mathbf{z}}_{i-1}) + \frac{n_i}{2}\text{Var}(\mathbf{W}_{i,-}\underline{\mathbf{z}}_{i-1})$$
$$= \frac{n_i}{2}(\text{Var}(\mathbf{W}_{i,+})\mathbb{E}(\overline{\mathbf{z}}_{i-1})^2$$
$$+ \text{Var}(\overline{\mathbf{z}}_{i-1})\mathbb{E}(\mathbf{W}_{i,+})^2$$
$$+ \text{Var}(\mathbf{W}_{i,-})\mathbb{E}(\underline{\mathbf{z}}_{i-1})^2 + \text{Var}(\underline{\mathbf{z}}_{i-1})\mathbb{E}(\mathbf{W}_{i,-})^2)$$
$$= \frac{\pi}{n_i}(1 - \frac{2}{\pi})\mathbb{E}(\overline{\mathbf{z}}_{i-1}^2) + \frac{2}{n_i}\text{Var}(\overline{\mathbf{z}}_{i-1})$$
$$+ \frac{\pi}{n_i}(1 - \frac{2}{\pi})\mathbb{E}(\underline{\mathbf{z}}_{i-1}^2) + \frac{2}{n_i}\text{Var}(\underline{\mathbf{z}}_{i-1}).$$

Note that $\mathbb{E}(\overline{\mathbf{z}}_i) \geq \mathbb{E}(\delta_i)$ and we have made $\mathbb{E}(\delta_i)$ stable in each layer. Thus $\text{Var}(\overline{\mathbf{h}}_i) \geq \frac{n_i}{2}\text{Var}(\mathbf{W}_{i,+})\mathbb{E}(\overline{\mathbf{z}}_{i-1})^2$ and will not vanish when the network goes deeper. Also note that $n_i > 1$ in neural networks, and therefore $\text{Var}(\overline{\mathbf{h}}_i)$ will not explode. The same analysis can also be applied to $\underline{\mathbf{h}}_i$.

However, when we use the IBP initialization, variance of the standard forward value $\mathbf{h}_i$ will be smaller than that of Xavier and Kaiming Initialization. Following the analysis in (He et al., 2015a), we have

$$\text{Var}(\mathbf{h}_i) = \frac{n_i}{2}\text{Var}(\mathbf{W}_i)\text{Var}(\mathbf{h}_{i-1}).$$

In IBP initialization, we have $\text{Var}(\mathbf{W}_i) = \frac{2\pi}{n_i^2}$, and the variance of $\mathbf{h}_i$ can become smaller after going through each affine layer. Therefore, as mentioned in Section B.1, simply adding IBP initialization may not finally improve the verified error, because it may harm the early warmup when $\epsilon$ is small and certified training is close to standard training. In this paper, in addition to IBP initialization, we further add regularizers to stabilize certified bounds and the balance of ReLU neuron states, while the variance is stabilized by fully adding BN. The effect of these parts of our proposed method is discussed in Section B.1.