# OpenReview forum: "Fast Certified Robust Training with Short Warmup"
_ICML.cc/2021/Workshop/AML — ICML 2021 Workshop AML Poster_

### Official Review · Reviewer_2xeD · 2021-06-19
**An engineering improvement of IBP training**

**Rating:** Accept
**Confidence:** 4

**Review:**

This paper proposed several tricks for accelerating the training of Interval Bound Propagation. The authors first list several observations that cause IBP to be slow. Then they use weight initialization and batch normalization to overcome the problem. The experimental results verified their observation and tricks.

---

### Decision · Program_Chairs · 2021-06-21

**Decision:**

Accept (Poster)

**Comment:**

This paper proposed several tricks for accelerating the training of Interval Bound Propagation. The presentation is clear and the experiments are solid.